

# An inversion of fine particulate matter (PM$_{2.5}$) mass concentrations based on the air quality index (AQI) during dust prone periods in Hotan oasis, Sinkiang

Ju Chunyan[1,2,+], Zhang Zili[3,+], Zhou Xu[4,+], He Qing[2]

[1]School of Management, Xinjiang Agricultural University, Urumqi, China

[2]Institute of Desert Meteorology, China Meteorological Administration, Urumqi, China

[3]Zhejiang Environment Monitoring Centre, Hangzhou, China

[4]Xinjiang Institute of Ecology and Geography, Chinese Academy of Sciences, Urumqi, China

+These authors contributed equally to this work.

*Correspondence to*: He Qing (qinghe@idm.cn)

**Abstract.** Ambient air pollution has been implicated as a major environmental problem in Urban development process. The objective of this publication is to analyse deeply the correlation coefficient of PM$_{2.5}$ and AOD and aerosol optical depth (AOD). Surface PM$_{2.5}$ observation data and AOD were investigated from March to June in 2015 and 2016. Hourly PM$_{2.5}$ data are sampled from air quality monitoring stations in Hotan oasis. The AOD data are derived from Terra and Aqua at 10 km resolution. The satellite passed the area at about 13: 30 AM and 15: 30 PM, respectively. By using the matched PM$_{2.5}$ and AOD data, the spatial and temporal distribution characteristics are discussed, and the correlation coefficient of PM$_{2.5}$ versus AOD are estimated. The results show that PM$_{2.5}$ mass concentration and AOD vary greatly in different pollution weather. This phenomenon may be associated with data collection time, and other meteorological factors. Regression analysis based on typical air pollution show subsection fitting effect is relatively good choice, and regression is relatively well in Hazardous and serious pollution weather. Fitting analysis is good for PM$_{2.5}$ in different level of air pollution, and sources and pollutants transmission have difference.

## 1. Introduction

Different levels of air pollution occur in cities, towns and villages with the rapid advance of urban construction and economic development. As the primary pollutant of air quality, the atmospheric particulate matter has been paid more and more attention. The dynamic monitoring play an important role in environmental protection and human social development(Seaton et al., 1995; Rd and Dockery, 2006; Chan and Yao, 2008; Pope et al., 2011; Lary et al., 2014). Particulate matter with aerodynamic diameters less than 2.5 mm (PM2.5 or PM$_{2.5}$) is related to mortality and morbidity outcomes for containing large amount of toxic and harmful substances(Pope et al.,2009; Fujii et al.,2001; U.S. Environmental Protection Agency (EPA)2004; Mordukhovich et al,2015).

At present, acquisition methods of PM$_{2.5}$ mass concentration are mainly from ground observation and remote sensing extraction. The applications of ground-level PM$_{2.5}$ monitoring method was limited due to their spatial coverage. The satellite data is now being used for monitoring atmospheric pollution, which become the important tool to air quality and pollution monitoring for its extensive spatial coverage(Al-Saadi et al.,2005; Prasad and Singh, 2007; Kaskaoutis et al., 2011, 2013;Chudnovsky et al.,





2014). One kind of extensive application method is study the particle dynamics of air pollution with aerosol optical thickness (AOD)(Lee et al., 2011,2016; Gupta, et al., 2013; Pozzer et al., 2015; You et al., 2016; Chu et al., 2016). Aerosol optical thickness is a comprehensive effect of the various characteristic parameters of aerosol particles, embodies the status of aerosols(Wang et al.,2005; Zhu et al.,2015). The
data of ground based Aerosol Robotic Network (AERONET)(Holben et al., 1998; Smirnov et al., 2000) have been used extensively to study aerosol optical properties, but the monitoring stations are scarce that there are no monitoring points in some areas. So the Ground observation data were needed for AOD estimation.

AOD estimation usually adopts dark target method, which is suitable for low surface reflectance and
stable ocean(Griggs, 1975), and dense vegetation(Kaufman & Sendra, 1988). However, the urban and desert regions have the high reflectivity. The visible light band of these areas has the highlight feature which makes it difficult to recognize the optical thickness of aerosol for satellite remote sensing data. The deep blue algorithm is the appropriate solution for the bright ground, and was verified in the aerosol inversion in the Sahara region(Tanre et al.,1988). The Moderate Resolution Imaging
Spectroradiometer(MODIS) onboard the Terra and Aqua satellites provide a daily global coverage, but its aerosol product (10 km Dark Target (DT)) is not suitable exposure estimates in urban areas. The MODIS aerosol product (MOD 04) uses a deep blue algorithm with a spatial resolution of 10 km, which provides basic data for the study of urban and desert areas(Sayer et al.,2015).

PM$_{2.5}$ mass concentration inversion is to build mathematical models based on data changes from
production of AOD(Schafer et al., 2008; Wallace et al., 2007; Gupta and Christopher, 2009), which chose the linear model a lot(Wang, 2003; Engel-Cox et al., 2004; Hutchison et al., 2005; Koelemeijer et al., 2006; Wang et al., 2010; Lee, 2011; Chudnovsky et al., 2013,2014; Toth et al., 2014; He et al., 2016; Karimian et al., 2016; Lin et al., 2016). However, the fitting effect is not ideal due to various factors(Naresh, et al., 2011). Wang and Christopher(2003) shows research that under certain conditions,
the ground PM$_{2.5}$ has a good correlation with the AOT of 550nm(R > 0.7), and the results have been verified (Gupta and Christopher, 2009). Pawan and Christopher(2009) studied the relationship of AOT and PM$_{2.5}$, the estimated value was 0.604, the slope was 0.37, and the intercept was 8.6. If thinking the meteorological factors, the value of R$^2$ increases to 0.683. Using MODIS deep blue AOD with land use regression to estimate PM$_{2.5}$ in California, the results showed high predictive power for PM$_{2.5}$
concentrations with R(2) = 0.66, and Deep Blue AOD in combination with land use regression can be particularly useful to generate spatially resolved PM$_{2.5}$ estimates(Lee et al.,2011). The other researches also proved deep blue AOD within a mixed effects model considerably could improve PM$_{2.5}$ prediction in high reflectance regions(Sorek-Hamer et al., 2015). But the facts showed based on AOT to invert PM$_{2.5}$, the fitting results are influenced by various factors, and the difference of PM$_{2.5}$ source is difficult
to distinguish.

Hotan oasis is located in the river alluvial fan plain oasis in the southern taklamakan desert. The Special geographical environment contribute to the serious air pollution in Hotan city. Large desert areas in the northern of Hotan city are the biggest source of dust weather affecting the PM measurements and air quality. This research selects Hotan oasis as the object to analyze the correlation between AOD and
PM$_{2.5}$. The data were obtained from ground air quality monitoring stations. The aim is to construct the suitable model for regional characteristics from March to June in 2015 and 2016. The results may provide a reference for inversion of PM$_{2.5}$ mass concentration based on AOD in the oasis.



## 2. Methodology

### 2.1 Study area

Hotan oasis (78°-81°83′ E,34°22´- 39°38′N) is located in the south of Xinjiang Uygur Autonomous region. The administrative division of research area covers one city and three counties (Hotan city, Hotan county, Moyu county and Lopu county), which located in the river alluvial fan plain oasis in the southern taklamakan desert, as shown in Figure 1. Spring is windy and sandy weather, the annual precipitation is sparse under Warm-temperate forest climate.

The sand and dust weather of Hotan oasis mainly appeared in April to October, and the period of high incidence was March to June. Sand and dust weather in meteorology can be divided into three types, dust, blowing sand and dust storms(He, et al., 2003). The dust weather statistics from March to June in 2015 was shown in Figure 2. If two types within a day, the more serious class will be take as the record. Dust storms occur in the calm winds, or low wind, and may be the unsinkable dust after dust storms or blowing sand. Due to wind power increasing, most of blowing sand weather partly changed into the weather from dust weather. The strong winds swept the dust from ground into the air to create the sandstorm, then the wind speed reduced and dust storms gradually turned into dust. From figure 2, it can be seen that the occurrence obvious effect on the atmosphere environment by various kinds of dust and dust weather in 2015 from March to June. The number of dust weather occurred most, which was conducive to the accumulation of $PM_{2.5}$.

### 2.2 Data and preprocessing

In this study, $PM_{2.5}$ concentration data were derived from the air quality publishing platform of China National Environmental Monitoring Centre (http://106.37.208.233:20035/). Currently, there are two air quality monitoring sites in research region. This paper focuses on urban studies, so we select the monitoring data in the central area of Hotan city. The time of monitoring was March to June in 2015 and 2016, respectively. The data exclude unvalued and outliers data.

The remote sensing data is MODIS aerosol product(MOD04 Level2 C051), which obtained from MOD04 (Terra) and MYD04 (Aqua) (http://ladsweb.nascom.nasa.gov/data/search.html). Data include 453 views, which take out the AOD value greater than 2 to eliminate impact of cloud detection(He,et al., 2010; Wang,et al., 2016). And then extract the non-zero data of the pixels centered on the monitoring station. The data from Terra is recorded as $AOD_T$, and Aqua data as $AOD_A$.

The time of passing study region of Terra is 13 or 14, and Aqua is 15 or 16. The $PM_{2.5}$ mass concentration data on the ground is hourly data, so it needs to match the two different data of Terra and Aqua. In this article, the data of Terra and Aqua are divided into two groups for comparison and analysis, and the matching $PM_{2.5}$ data is chosen as the average value of the satellite transit time in 1 hour.

### 2.3 Methods

In order to investigate the changing characteristics of $PM_{2.5}$ mass concentration and AOD, the data were statistically analyzed. Descriptive statistics, frequencies were performed. Associations of $PM_{2.5}$ to AOD based on AQI were investigated with the Pearson's correlation coefficient. The Linear regression analysis was applied to examine the significant correlation coefficients of $PM_{2.5}$ and AOD. Statistical analyses were performed with the SPSS.

According to local weather characteristics, and references to the $PM_{2.5}$ concentrations in US Mission China website(URL:http://stateair.net) and along with health/activity warnings as indicated, the values



of AQI was shown in table 1 (Mintz,2013; Zheng et al., 2014).

## 3 Result

### 3.1 Variation features of PM2.5 mass concentration

According to local weather characteristics, the monitoring data was divided into seven classes(Figure 3)(Zheng et al., 2014). When AQI reached 500 in 2015, data collected from Terra and Aqua was 27.5 percent and 26.39 percent respectively. The data of Terra and Aqua data was 13.89 and 16.67 in 2016. The number of days has decreased when AQI reached 500 in 2016, but pollution weather of unhealthy for sensitive groups has increased than 2015 (AQI value are 101 to 150). The days of unhealthy(151-200) has increased than 2015 in Aqua. The air pollution in this area is serious, and the obviously difference among different periods.

In this article, the $PM_{2.5}$ mass concentration were statistical analyzed firstly in Figure 4. The figure shows differences of $PM_{2.5}$ mass concentration in different air quality pollution in 2015 and 2016. The median of $PM_{2.5}$ concentration is not obvious in better air quality. As the degree of air pollution deepen, the median overall appears to rise. Median are highest as AQI equal to 500. The median was lower in moderately polluted than that of light pollution in 2015. Comparing the data sets in Figure 4, it can be found that the median fluctuation of $PM_{2.5}$ mass concentration is more obvious in 2015.

In order to research the change of $PM_{2.5}$ mass concentration, the standard deviation is introduced, which indicates discrete degree of data in statistics. The range of $PM_{2.5}$ mass concentration are relatively consistent and stable in fine weather. The extreme value of $PM_{2.5}$ mass concentration varies significantly under polluted weather conditions.

Among them, $PM_{2.5}$ mass concentration was significantly higher in the morning than in the afternoon, and the range increased when air pollution increased. When AQI reached 500, the range was higher than that of the same group. The $PM_{2.5}$ mass concentration in the morning was more obvious than afternoon period, especially range reaching the maximum in 2016. And $PM_{2.5}$ mass concentration varies greatly in different pollution weather in the morning. When AQI reaches 500, the $PM_{2.5}$ mass concentration reaches the maximum. This phenomenon may be associated with data collection time, and other meteorological factors. Data collection time from Terra is 13 or 14 , and Aqua data is 15 to 16 points. Furthermore, the emergence of dust weather in urban area and the active degree of human activity can affect the $PM_{2.5}$ mass concentration.

### 3.2 Characteristics Analysis of the of AOD

Basic statistical analysis of AOD matching with $PM_{2.5}$ as shown in Figure 5. The figures show that AOD has the significant trend in different levels of pollution. The median of AOD is relatively high when AQI reaches 500(Ⅶ) or hazardous pollution weather(Ⅵ). The median changes of AOD were relatively small in 2015, and the great difference in different time periods in 2016. In the morning, the data was the highest in pollution weather of hazardous (Ⅵ).

Range changes of AOD data also have obvious difference. Range is maximum when AQI reaches 500 or pollution weather of hazardous(Ⅵ) in 2015. Range is maximum during serious pollution weather in morning in 2016, and the maximum of range appeared when AQI reaches 500(Ⅶ) in afternoon. In addition, the comparison of the statistical parameter showed that certain difference between $PM_{2.5}$ and AOD, this may be impact of data collection, such as interpolation of the AOD has the certain error





exists.

### 3.3 Correlation analysis based on air pollution index (AQI)

Based on the above analysis, the $PM_{2.5}$ mass concentration and AOD show the different changes in different levels of pollution. The one method of regression analysis will ignore the detailed difference,which result to inaccurate for the whole data. We adapt the new method to explore the deeply relationship between $PM_{2.5}$ and AOD.

Relationship analysis was used in the total sample points respectively according to AQI degree to analyze the relationship between $PM_{2.5}$ and AOD. Results showed detailed relationship between AOD and $PM_{2.5}$ in Hotan oasis in different pollutant environment(Table 2). Pearson value of $AOD_T$ and $PM_{2.5}$ is 0.525, $AOD_A$ and $PM_{2.5}$ is 0.695. Value of Sig. (2-tailed) is 0.000, all results pass significance test. Table 2 shows the different correlation between AOD and $PM_{2.5}$. There is a high correlation of $PM_{2.5}$ and AOD in the pollution weather and results pass significance test than others. As the increase of air pollution levels, the correlation increased in some period. Correlation value reached highest(0.961) in morning in 2016. So high correlation between AOD and $PM_{2.5}$, but has the difference. It show obvious difference in different weather conditions for AOD and $PM_{2.5}$.

### 3.4 Regression analysis based on typical air pollution

Based on the results of Table 1, the regression analysis of matching AOD and $PM_{2.5}$ data was selected which pass significance test (Table 3). The results show subsection fitting effect is better in 2015 and the $R^2$ value is higher than the total sample. Especially in 2015, AQI has two stages: 0-100, 301-499, and the results are obviously improved. When AQI reaches 500(Ⅶ), the fitting effect is not ideal, which is related to the type of aeolian sand disaster. The dust weather conditions are not conducive gathering for $PM_{2.5}$. In 2016, the fitting effect was different and the $R^2$ value was larger. The correlation was higher and the $R^2$ value decreased. In general, the result in morning is better than afternoon. Piecewise fitting is relatively good choice that regression is relatively well in hazardous and serious pollution weather.

### 4 Discussion

In this paper we use the subsection fitting effect to fit the AOD and $PM_{2.5}$ data. The main goal was to study the difference of relationship between $PM_{2.5}$ and AOD to improve our ability to know quantitatively spatial relationship patterns of $PM_{2.5}$ and AOD. Importantly, we have shown that $PM_{2.5}$ prediction from AOD based on AQI, which improves accuracy further with subsection fitting effect analysis. We have shown that using the subsection fitting effect we obtain better predictive power spatially.

From the analysis of relationship between $PM_{2.5}$ and AOD, it is of high importance to have information about the spatial correlation of $PM_{2.5}$ and AOD in different pollutant levels. In the previous studies, linear model were used to predict $PM_{2.5}$ concentrations(Wang, 2003; Engel-Cox et al., 2004; Hutchison et al., 2005; Koelemeijer et al., 2006; Wang et al., 2010; Lee, 2011; Chudnovsky et al., 2013; Toth et al., 2014; He et al., 2016; Karimian et al., 2016; Lin et al., 2016). Pawan and Christopher(2009) studied the relationship of AOT and $PM_{2.5}$, the estimated value was 0.604, the slope was 0.37, and the intercept was 8.6. If thinking the meteorological factors, the value of $R^2$ increases to 0.683. In contrast, our study



showed high accuracy in some polluted weather with piecewise fitting.

The Hotan oasis is located in the southern taklamakan, which affected by the wind and sand movement in the desert. The air quality is very serious. By analyze and compare the statistical parameter(Figure 4), $PM_{2.5}$ mass concentration changes complex with obvious difference in different pollution levels.

Acquisition time of data has the obvious difference in the morning and afternoon. This is due to the complicated sources of $PM_{2.5}$, including desert and human activities.

The change of atmospheric particulate concentration in spring and summer in Hotan oasis is obviously affected by the dust weather(Liu, et al.,2011), and the wind speed is a very important factor. The particle size of $PM_{2.5}$ is relatively small, the greater the wind speed affect its gathered. So wind speed in dust

weather environments conducive to gathered for $PM_{2.5}$ , and left in the air for a long time, the impact is also bigger for air quality .

Aerosol optical thickness is the synthetical effects of the aerosol particles. Correlation analysis results show significant correlation of AOD and $PM_{2.5}$. But under the different air pollution levels, related degree also exist obvious differences. Fitting analysis is good for $PM_{2.5}$ in different level of air pollution,

and sources and pollutants transmission have difference. The more research need to do according to specific conditions.

## 5 Conclusion

Based on MODIS AOD product and ground hourly data of matching $PM_{2.5}$ from March to June in 2015 and 2016, the article analyze correlation of MODIS aerosol optical depth (AOD) and PM2.5. Statistical

characteristics and regression analysis were done in the Hotan oasis, the results as show below.

$PM_{2.5}$ concentrations data vary in different levels of pollution, it has relationship with the source of the particulate matter and transmission route, sandstorm weather had the greatest impact on $PM_{2.5}$ from the taklamakan desert.

The statistical parameters of AOD show that the median and rang of data are both higher in hazardous

pollution. The rang of $AOD_T$ is larger than that of $AOD_A$ in the morning. The collection time of $AOD_T$ is about 13 to 14, while $AOD_A$ is 15 to 16, all of which have an impact on AOD.

AOD was correlated with matching PM2.5 in different air pollution levels, but there are differences in the correlation. Through correlation and regression analysis, there exists significant difference for $PM_{2.5}$ in different level of air pollution condition. Phased fitting effect is better, which affected by the sources of

pollutants, route of transmission route.

The inversion of aerosol optical thickness and the determination of $PM_{2.5}$ are affected by climate conditions (such as temperature, humidity, wind, visibility), geographic information (topography and land use) and the influence of factors such as season, boundary layer height. The uncertainty of parameters also influence the accuracy of the results. Based on the characteristics of various factors and

the availability of data, the follow-up work needs to explore the influence factors of various factors on the inversion of aerosol optical thickness and the restriction of $PM_{2.5}$. Meanwhile, the AOD data in this article is used to verify new method's feasibility with the AOD data,which was derived from the Terra and Aqua satellite at 10 km resolution. The further study will use high resolution data to do analysis deeply.



**Acknowledgement**

This research was supported by National Natural Science Foundation of China under Grant 41375163 and 41641045. Thanks for the MODIS data products provided by NASA LAADS and reviewer.

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



**Table 1: Air quality index range**

| Category | Degree of air quality | AQI value |
|---|---|---|
| Good | I | 0-50 |
| Moderate | II | 51-100 |
| Unhealthy for Sensitive Groups | III | 101-150 |
| Unhealthy | IV | 151-200 |
| Very Unhealthy | V | 201-300 |
| Hazardous | VI | 300-499 |
| Exceed standard | VII | 500 |



**Table 2 : Pearson correlation matrix in the March-June**

| Air pollution index(AQI) | Item | 2015 | | 2016 | |
| --- | --- | --- | --- | --- | --- |
| | | MOD | MYD | MOD | MYD |
| 0－100 | Pearson Correlation | -.359 | -.009 | .050 | .275 |
| | Sig. (2-tailed) | .343 | .984 | .870 | .286 |
| | N | 9 | 8 | 13 | 17 |
| 101－150 | Pearson Correlation | .334 | -.125 | .406 | .135 |
| | Sig. (2-tailed) | .223 | .685 | .085 | .632 |
| | N | 15 | 13 | 19 | 15 |
| 151－200 | Pearson Correlation | .416 | -.115 | .320 | .673** |
| | Sig. (2-tailed) | .232 | .737 | .401 | .008 |
| | N | 10 | 11 | 9 | 14 |
| 201－300 | Pearson Correlation | .877** | .781* | -.072 | .499 |
| | Sig. (2-tailed) | .002 | .013 | .855 | .314 |
| | N | 9 | 9 | 9 | 6 |
| 301-499 | Pearson Correlation | .431 | .870** | .796** | .641 |
| | Sig. (2-tailed) | .214 | .001 | .003 | .087 |
| | N | 10 | 10 | 11 | 8 |
| 500 | Pearson Correlation | .319 | .588* | .961** | .623* |
| | Sig. (2-tailed) | .159 | .008 | .000 | .030 |
| | N | 21 | 19 | 10 | 12 |

\* The correlation was significantly at 0.05 level(2-tailed)

\*\* The correlation was significantly at 0.01 level(2-tailed)



**Table 3: Regression parameter of MOD in the March-June**

| Model | Total sample | | | | 201−300 | | | |
|---|---|---|---|---|---|---|---|---|
| 2015 | B | Std. Error | Sig. | R² | B | Std. Error | Sig. | R² |
| Constant | .081 | .079 | .310 | .276 | -.147 | .109 | .219 | .769 |
| PM₂.₅ | .003 | .001 | .000 | | .005 | .001 | .002 | |

| Model | Total sample | | | | 301-499 | | | | 500 | | | |
|---|---|---|---|---|---|---|---|---|---|---|---|---|
| 2016 | B | Std. Error | Sig. | R² | B | Std. Error | Sig. | R² | B | Std. Error | Sig. | R² |
| Constant | .059 | .060 | .326 | .65 | -.014 | .182 | .942 | .757 | -.168 | .305 | .588 | .346 |
| PM₂.₅ | .003 | .000 | .000 | | .005 | .001 | .001 | | .005 | .002 | .008 | |



**Table 4: Regression parameter of MYD in the March-June**

| Model | Total sample | | | | 301-499 | | | | 500 | | | |
|---|---|---|---|---|---|---|---|---|---|---|---|---|
| 2015 | B | Std. Error | Sig. | $R^2$ | B | Std. Error | Sig. | $R^2$ | B | Std. Error | Sig. | $R^2$ |
| Constant | -.130 | .085 | .131 | .484 | .120 | .352 | .742 | .634 | -.158 | .137 | .282 | .924 |
| $PM_{2.5}$ | .006 | .001 | .000 | | .005 | .001 | .003 | | .003 | .000 | .000 | |
| Model | Total sample | | | | 151-200 | | | | 500 | | | |
| 2016 | B | Std. Error | Sig. | $R^2$ | B | Std. Error | Sig. | $R^2$ | B | Std. Error | Sig. | $R^2$ |
| Constant | .035 | .064 | .586 | .582 | -.320 | .240 | .207 | .453 | .380 | .425 | .392 | .388 |
| $PM_{2.5}$ | .004 | .000 | .000 | | .011 | .004 | .008 | | .003 | .001 | .030 | |




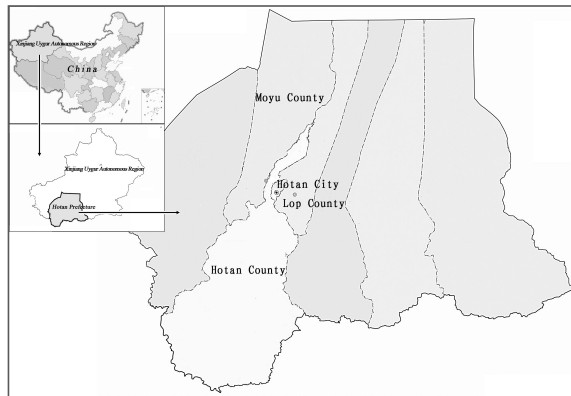

**Figure 1: Geographical location of the research region**



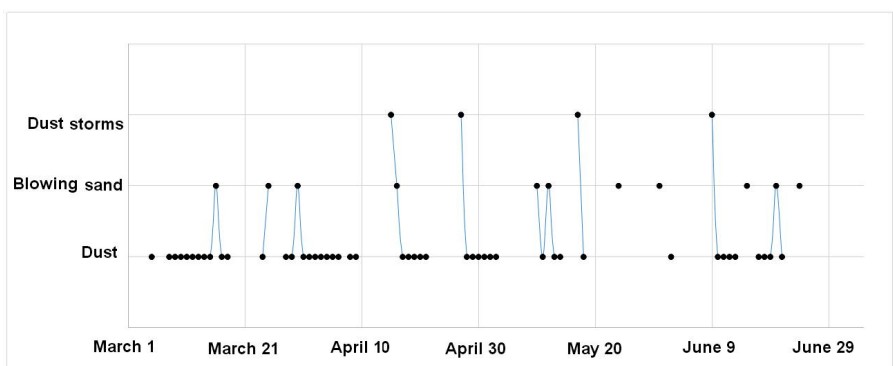

Figure 2: Graph of sand and dust weather distribution in the March-June period of 2015

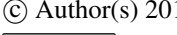



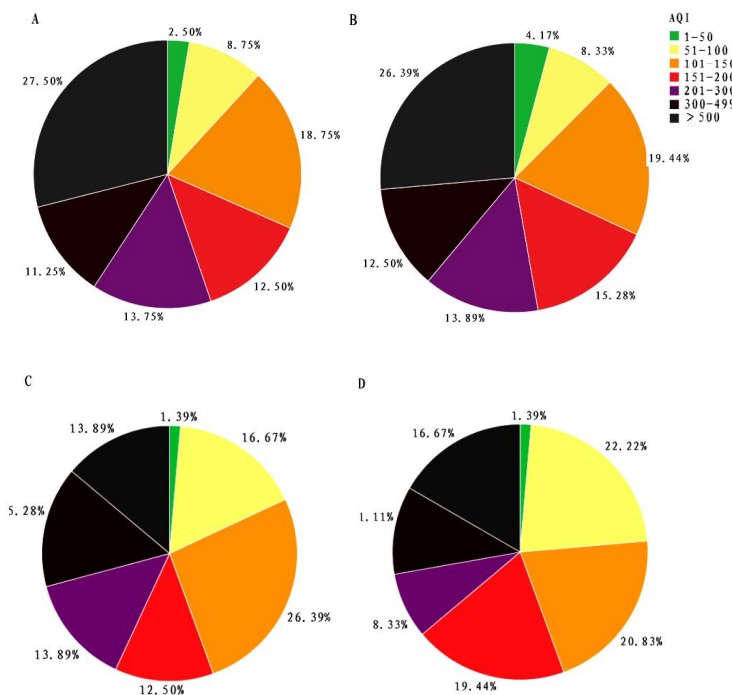

**Figure 3: Graph of Air quality distribution in the March–June period（A.2015 年 AOD$_T$ B.2015 年 AOD$_A$ C.2016 年 AOD$_T$ D.2016 年 AOD$_A$）**





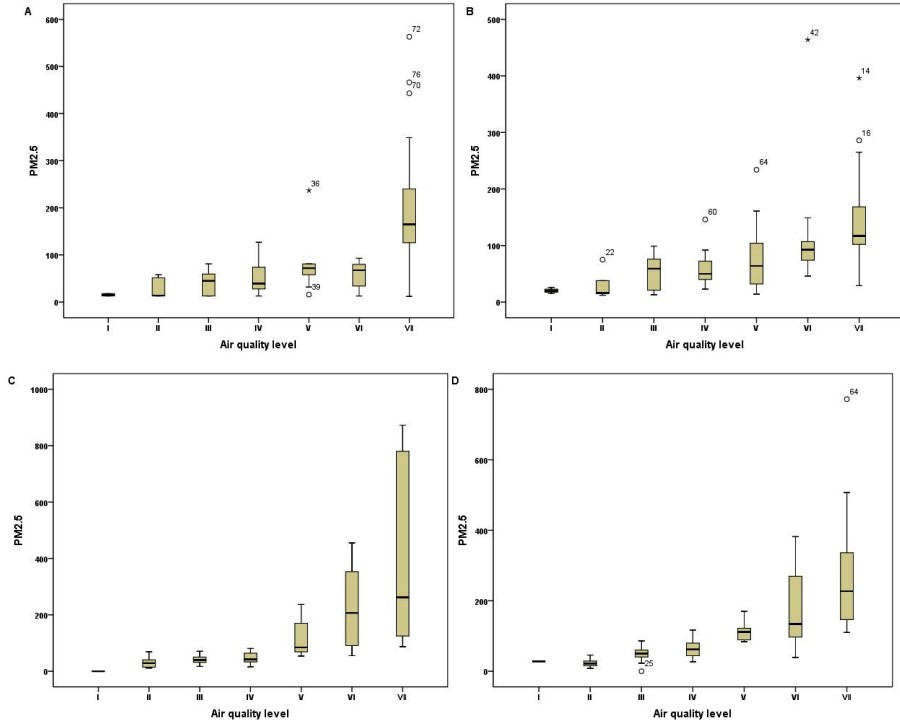

Figure 4: Graph of PM$_{2.5}$ concentration distribution in the March-June (A.AOD$_T$ in 2015 B.AOD$_A$ in 2015 C.AOD$_T$ in 2016 D.AOD$_A$ in 2016)





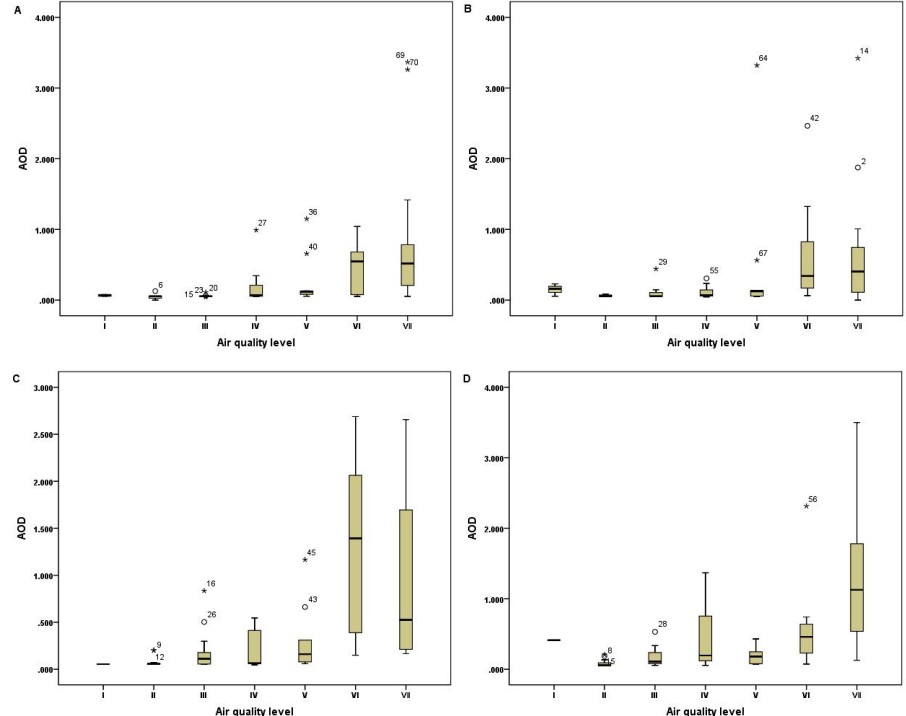

**Figure 5: Graph of AOD concentration distribution in the March-June (A.AOD$_T$ in 2015 B.AOD$_A$ in 2015 C.AOD$_T$ in 2016   D.AOD$_A$ in 2016)**