# Peer review of "An inversion of fine particulate matter ( $PM_{2.5}$ ) mass concentrations based on the air quality index (AQI) during dust prone periods in Hotan oasis, Sinkiang"

_Natural Hazards and Earth System Sciences, 2017_

## Referee Comment (RC1) · Anonymous Referee #1 · 27 Jun 2018

The paper entitled "An inversion of fine particulate matter (PM2.5) mass concentrations based on the air quality index (AQI) during dust prone periods in Hotan oasis, Sinkiang" by Chunyan, I., Zili, Z., Xu, Z. and Qing, H. intends to provide a reference for the inversion of PM2.5 mass concentrations from satellite-derived aerosol optical depth (AOD) data in the Hotan oasis area (China) during periods affected by sand and dust events, using a classification based on the air quality index (AQI). To this aim, PM2.5 hourly concentrations recorded by air quality stations located in the area under study and AOD values derived from the Moderate Resolution Imaging Spectroradiometer

(MODIS) aboard the Terra and Aqua satellites during their overpasses over this area in the period March-June 2015/2016 were considered. Firstly, the PM2.5 concentrations and AOD values were grouped on the basis of different air quality conditions - assessed using the air quality index (AQI) - and a discussion of their statistics was provided. Then, the correlations between PM2.5 and AOD data as a function of the AQI values were investigated. Finally, the linear regression analysis was applied to find the fitting relationship between PM2.5 and AOD by considering the total dataset and by dividing it using the AQI values and a discussion of the results obtained was provided.

GENERAL COMMENTS

Although the manuscript addresses a relevant scientific topic such as the atmospheric pollution and is within the scope of NHESS, it does not give a substantial contribution to the understanding of this hazard and its consequences. The scientific approach and the applied method are very weak. The results and their discussion are very confused and there is not sufficient evidence to support the interpretations of the conclusions given. The reference section is also very inaccurate. In general, the work appears of local interest only and I cannot see original and new findings that the scientific community could learn from this study. Moreover, even though I am not a native English speaker, I think that there are a lot of problems with grammar and the use of English that worsen the understanding of the work. To end with, I do not recommend the publication of this paper in its current form on NHESS journal. In the following, some specific comments that supported my decision. Moreover, minor technical comments are also provided.

SPECIFIC COMMENTS

Page 3, line 8: What is the methodology used to identify the sand and dust events?

Page 3, line 9: What is the criterion used to classify the sand and dust events into the three types considered? The article cited as a reference is not easily accessible.

Page 3, line 11: Why does Figure 2 report the number of dust, blowing sand and dust storm events for the period March-June 2015 only? What happens in 2016? What do the Authors mean when they write "If two types within a day, the more serious class will be taken as the record"? What does "serious class" mean? How do they define a more serious class than the other? Which parameter do they use?

Page 3, line 37: The Authors introduce the AQI without providing a minimal description, assuming that everyone knows what it is. Moreover, they do not specify that it is referred to PM2.5 only.

Page 4, line 5: It is not clear the scientific content of Figure 3. Why do the Authors consider of scientific interest to divide the AOD data into 7 classes according to the AQI? What is the result they want to highlight?

Page 4, lines 4-10: Firstly, it is not clear the scientific content of Figure 3. Then, why do the Authors discuss the AOD in the paragraph dedicated to the PM2.5 data discussion? Why do they consider of scientific interest to divide the AOD data into 7 classes according to the AQI? What is the result they want to highlight? Moreover, the AQI should be based on PM2.5 daily concentrations and AOD data are referred to a specific hour. How do the Authors manage this?

Page 4, lines 11-16: The discussion of the results shown in Figure 4 is very weak.

Page 4, line 18: What does "discrete degree of data" mean?

Page 4, lines 21-22: What do the Author mean when they write: "Among them, PM2.5 mass concentration was significantly higher in the morning than in the afternoon, and the range increased when air pollution increased"? I was not able to find the PM2.5 values supporting this statement.

Page 4, line 32: How do the Authors establish that the AOD has a significant trend?

Page 4, lines 31-40: How do the Authors explain the variability of PM2.5 mass concentrations and AOD values as a function of the AQI classes?

Page 5, lines 9-10: The Pearson's correlation values between PM2.5 concentrations and AOD data cannot be found in Table 2. It is quite difficult for a reader to understand that they are referred to the entire dataset and not to its classification as a function of the AQI, especially if the correlation values are reported immediately after mentioning Table 2 in the text.

Page 5, lines 11-15: The results show that the correlation between PM2.5 and AOD increases with the increase of the AQI values. However, the Authors limit themselves to reporting the values shown in Table 2 without providing a scientific explanation only describe the table without providing any scientific interpretation of the results found.

Page 5, lines 16-24: The regression analysis section should be the main part of the article as the Authors aim to provide a reference for the inversion of PM2.5 mass concentrations from satellite-derived AOD data using a classification based on the AQI. This section is incomplete and the discussion, as well as being difficult to understand for the misuse of English, is very weak and limited to a poor description of the data reported in the corresponding tables (i.e. Table 3 and Table 4. Moreover, the latter is never recalled in the text) that are the output of the statistical program used (i.e. SPSS). In addition, although the fitting obtained by subdividing the dataset according to the AQI is better than the one obtained considering the entire dataset, it is not clear what this improvement is linked to. The Authors do not provide a robust explanation.

Page 5, lines 26-31: The Authors say that "the main goal was to study the difference of relationship between PM2.5 and AOD to improve our ability to know quantitatively spatial relationship patterns of PM2.5and AOD". No spatial relationship has been found.

Discussion section: Most of the text of this section is a useless repetition of what has already been written and does not add anything to the discussion of the results. This section should be completely rewritten or deleted.

Reference section: A careful review of the reference section is needed. For example, as reported in the general comments, the same article is reported twice differently (e.g.

[Figure]

page 7, lines 30-32 and page 9, lines 4-6). Alphabetical order is not always respected (e.g. see the reference reported at page 9, lines 19-21) and some articles are mentioned in the text but not in the references section (e.g. Mordukhovich et al.,2015). Moreover, some articles are not accessible by fellow scientists.

MINOR TECHNICAL COMMENTS

Page 1, line 16: The hours are not correctly reported. Moreover, it is not clear if it is local time or UTC. It would be better to express the time in UTC.

Page 2, line 2: Aerosol optical thickness is generally reported as AOT. To avoid confusion, it would be better to use "depth" instead of "thickness" so as to use the acronym AOD throughout the manuscript.

Page 3, line 21: The website reported as http://106.37.208.233:20035/ is not accessible.

Page 3, line 30: The Authors report: "The time of passing study region of Terra is 13 or 14, and Aqua is 15 or 16". As said in a previous comment, it would be better to express the time in UTC. Moreover, the time of the overpasses over the region under study should be checked.

Page 12, Table 2: It would be better to use AODT and AODA in place of MOD and MYD.

Page 14, Table 4: Table 4 is not mentioned in the text.

Page 17, Figure 3: The figure reports Chinese symbols.

Page 18, Figure 4: The figure does not report the PM2.5 unit of measurement. Moreover, it would be better to provide a legend of the symbols reported in the figure (e.g. The line in the centre of the box represents the median value, lower and upper boundaries for each box are . . .. and the whiskies encompass . . . times the range of the box. Finally, asterisks and empty circles correspond to . . ...).

Page 19, Figure 5: As reported in the previous comment, it would be better to provide a legend of the symbols shown in the figure.

---

## Referee Comment (RC2) · Anonymous Referee #2 · 22 Jul 2018

Review of the manuscript entitled "An inversion of fine particulate matter (PM2.5) mass concentrations based on the air quality index (AQI) during dust prone periods in Hotan oasis, Sinkiang" by, Ju Chunyan, Zhang Zili, Zhou Xu, He Qing, No.: nhess-2017-341

This manuscript deals with the investigation of the relationship, in terms of correlation, between the PM2.5 concentrations from ground based measurements, and the satellite-derived Aerosol Optical depth (AOD) in the context of an ultimate goal to provide the frame of the PM2.5 mass concentrations inversion from satellite AOD data in the broader area of Hotan oasis (China) during time periods of the year affected by

dust events.

The analysis is based on hourly concentrations of PM2.5 measured by two stations of an air quality monitoring network and AOD values obtained from both, MODIS-Terra and MODIS-Aqua databases, retrieved with the Deep Blue algorithm at a spatial resolution of 10 Km. At first a descriptive statistics in terms of AOD frequency of occurrence and AOD and PM2.5 median and variability as a function of the Air Quality Index (AQI) is presented. In the next, the correlation between PM2.5 and AOD is examined for each class of AQI, whereas the linear regression analysis is applied to the whole dataset of PM2.5 and AOD and to subsets corresponding to different AQI classes as well, in order to determine the fitting relationship between them.

Although the submitted work presents a local interest, the subject treated is interest due to multiple implications of particulate matters in various environmental issues (e.g. climate studies, air quality issues, human health effects, . . .). Therefore, any additional information that can contribute to decrease the great uncertainty characterising the aerosol concentrations, optical and physical properties and their effects should be taken into account. A factor significantly contributing to high temporal and spatial variability of aerosol properties and thus to their high uncertainty, is the sporadic nature of some aerosol sources such as the dust emitted and transported from arid and semi-arid regions, particularly when it manifests as episode. Strong dust storm events can be a serious hazard mainly to human health and to ground and air transportation because of the increased load of particulate matter in air.

In this framework the subject treated by the submitted manuscript is relevant to the topics of NHESS journal. However, the paper is confused and not well written. It lacks novel concepts, tools and/or results and the applied methods are basic. Moreover, the discussion of the results and their interpretation is poor. They do not offer any significant contribution to the understanding of the factors determining the PM2.5 levels in the study area. One of the weaknesses of this manuscript is the use of the English that many times makes difficult the understanding of the content. For that reasons I

believe that the present manuscript is not suitable for publication in NHESS.

Some comments are presented below: • Page 3, lines 8-9: It is written "The sand and dust weather of Hotan oasis mainly appeared in April to October, and the period of high incidence was March to June.". Firstly, I think the first half of the sentence should be modified by replacing the April with March in order to be consistent with the second one and with the whole work. Secondly, in combination with the following sentence in lines 9-10 "Sand and dust weather in meteorology can be divided into three types, dust, blowing sand and dust storms (He, et al., 2003).", what are the methodology and the criteria used to define the occurrence of sand and dust events and to classify them in one of the three considered types: dust, blowing sand and dust storm? The discussion following this sentence underlines the role of the wind intensity but the last sentence of the paragraph along with figure 2, imply also the use of PM2.5 levels. • Page 3, lines 11: it is written "If two types within a day, the more serious class will be take as the record", what authors mean by the "serious class"? Do they mean the class with the most severe events? • Though great part of the statistical analysis of PM2.5 and AOD as well as the study of their correlation is conducted as a function of the AQI, authors do not describe this index. They just provide two references and they do not clarify if PM2.5 are accounted for in its calculation. Additionally, they should explain their choice to carry out the analysis in relation to AQI classes and particularly to divide AOD with respect to AQI classes. What the scientific interest is and what they expected to reveal? • Authors, throughout the discussion mention often the local weather characteristics but they do not give any information about the prevailing weather during the study period. For instance, in page 4 (line 4) they state "According to local weather characteristics, the monitoring data was divided into seven classes". The AOD data are grouped into seven classes with respect to AQI values, and obviously the AQI classes are related to prevailing local weather conditions but authors do not give any information on the weather corresponding to each class. • Page 4, line 33; it is cited "The figures show that AOD has the significant trend in different levels of pollution", did authors examine statistically the significance of this "trend"? It would

better to use the expression "dependence" instead of "trend". • The presentation of the results is rather descriptive while there isn't any interpretation or scientific explanation. For instance what factors determine the variation of PM2.5 and AOD among the AQI classes? Similarly, based on their analysis (page 5, lines 11-13), authors found that the correlation between PM2.5 and AOD increases with polluted condition (increased AQI values) but they do not address any explanation for this behavior. • Many times in the discussion authors mention differences in the PM2.5 levels within the day and specifically between morning and afternoon (e.g. page 4, lines 21-25: "Among them, PM2.5 mass concentration was significantly higher in the morning than in the afternoon, and the range increased when air pollution increased.", "The PM2.5 mass concentration in the morning was more obvious than afternoon period, especially range reaching the maximum in 2016. And PM2.5 mass concentration varies greatly in 25 different pollution weather in the morning." or page 4, lines 34-39 "In the morning, the data was the highest in pollution weather of hazardous (VI).", "Range is maximum during serious pollution weather in morning in 2016, and the maximum of range appeared when AQI reaches 500(VII) in afternoon." and page 5, lines 13-14: "Correlation value reached highest(0.961) in morning in 2016.", page 5, lines 22-23: "The correlation was higher and the R2 value decreased. In general, the result in morning is better than afternoon.", page 6 line 5: "Acquisition time of data has the obvious difference in the morning and afternoon.", page 6, line 25: "The rang of AODT is larger than that of AODA in the morning."). Additionally, they stated that the PM2.5 values used in this work are hourly averages around the time of satellite overpass in order to match them with the AOD values (page 3, lines 32-33: "In this article, the data of Terra and Aqua are divided into two groups for comparison and analysis, and the matching PM2.5 data is chosen as the average value of the satellite transit time in 1 hour") and in page 3, line 30 it is cited regarding the satellite time overpass that "The time of passing study region of Terra is 13 or 14, and Aqua is 15 or 16.". First, it should be made clear if this time is local time (LT) or universal (UTC). Second, in any case I can't see how PM2.5 values referred to satellite overpass time allow examining differences between morning

and afternoon time. If the time is local, values matched to Terra overpass correspond to midday (noon) time period with values matched to Aqua to correspond to the afternoon. If it is UTC time (-8 hours) values correspond to early in the morning and morning time period. • In the concluded section it is reported "The rang of AODT is larger than that of AODA in the morning. The collection time of AODT is about 13 to 14, while AODA is 15 to 16, all of which have an impact on AOD." but in the results analysis authors do not discuss the possible factors that induce this difference in a time space of two hours. Is it the development of the boundary layer (if the overpass time is the local time) or it is related to anthropogenic activities or an established local atmospheric circulation affects the dust transport? • Authors point out the role of meteorological conditions and particularly of the wind in influencing the PM2.5 levels (e.g. page 6, lines 7-9: "The change of atmospheric particulate concentration in spring and summer in Hotan oasis is obviously affected by the dust weather (Liu, et al.,2011), and the wind speed is a very important factor. The particle size of PM2.5 is relatively small, the greater the wind speed affect its gathered."), but they do not consider this factor in their analysis. • Though authors state in the text (page 2, lines 40-42) "The aim is to construct the suitable model for regional characteristics from March to June in 2015 and 2016. The results may provide a reference for inversion of PM2.5 mass concentration based on AOD in the oasis.", the section treats the regression analysis between PM2.5 and AOD is too short and the discussion is really poor. In their analysis, better correlation and fitting were obtained for sub-datasets created based on the AQI classes compared to the entire dataset, however authors do not interpret or explain the possible reasons of this improvement. • Page 5, lines 26-28: it is stated that "The main goal was to study the difference of relationship between PM2.5 and AOD to improve our ability to know quantitatively spatial relationship patterns of PM2.5and AOD.", however the spatial variability of the PM2.5 and AOD correlation is not investigated or discussed in the manuscript. • Discussion section: the content of this section doesn't interpret the findings of this work neither attempts a comparison with other studies. Actually, it is a repetition of the introductory section. • Concluding section: conclusions are rather

qualitative than quantitative. • Finally, English should drastically improved

Some minor comments • Page 2, line 2: aerosol optical thickness is denoted as AOT and not AOD. AOD is used to express the aerosol optical depth. • Page 2, lines 110-12: the sentence "The visible light band of these areas has the highlight feature which makes it difficult to recognize the optical thickness of aerosol for satellite remote sensing data" is not clear. Please rewrite it paying attention to English. • Page 3 in the Data section (2.2): please note at which wavelengths is referred the AOD used. I suppose that it is the AOD at 550 nm. • Page 14, Table 4: Table 4 is not discussed in the text. • Page 17, Figure 3: The figure's caption contains some Chinese symbols. Do they have any meaning? • Page 18, Figure 4: The figure and the figure's caption as well miss the units of the PM2.5 concentration. • Special care should be given to references.

Please also note the supplement to this comment:
https://www.nat-hazards-earth-syst-sci-discuss.net/nhess-2017-341/nhess-2017-341-RC2-supplement.pdf